# "Studying How to Efficiently and Effectively Guide Models with Explanations" - A Reproducibility Study

**Adrian Sauter**                                    *adrian.a.sauter@student.uva.nl*
*University of Amsterdam*

**Milan Miletić**                                    *milan.miletic@student.uva.nl*
*University of Amsterdam*

**Ryan Ott**                                    *ryan.ott@student.uva.nl*
*University of Amsterdam*

**Rohith Saai Pemmasani Prabakaran**      *rohith.saai.pemmasani.prabakaran@student.uva.nl*
*University of Amsterdam*

**Reviewed on OpenReview:** *https://openreview.net/forum?id=9ZzASCVhDF*

## Abstract

Model guidance describes the approach of regularizing the explanations of a deep neural network model towards highlighting the correct features to ensure that the model is "right for the right reasons". Rao et al. (2023) conducted an in-depth evaluation of effective and efficient model guidance for object classification across various loss functions, attributions methods, models, and 'guidance depths' to study the effectiveness of different methods. Our work aims to (1) reproduce the main results obtained by Rao et al. (2023), and (2) propose several extensions to their research. We conclude that the major part of the original work is reproducible, with certain minor exceptions, which we discuss in this paper. In our extended work, we point to an issue with the Energy Pointing Game (EPG) metric used for evaluation and propose an extension for increasing its robustness. In addition, we observe the EPG metric's predisposition towards favoring larger bounding boxes, a bias we address by incorporating a corrective penalty term into the original Energy loss function. Furthermore, we revisit the feasibility of using segmentation masks in light of the original study's finding that minimal annotated data can significantly boost model performance. Our findings suggests that Energy loss inherently guides models to on-object features without the requirement for segmentation masks. Finally, we explore the role of contextual information in object detection and, contrary to the assumption that focusing solely on object-specific features suffices for accurate classification, our findings suggest the importance of contextual cues in certain scenarios.
Code available at: https://github.com/ryan-ott/model-guidance-reproducibility.

## 1 Introduction

With the successful integration of artificial intelligence (AI) systems across various sectors, deep neural networks (DNNs) are increasingly involved in consequential decision-making processes. The significance of the decisions in these critical sectors necessitates the need for interpretable AI systems for the purpose of regulatory compliance, ethical considerations and gaining user trust. Compared to traditional machine learning algorithms such as decision trees and linear regression models, which are inherently interpretable due to their simple architecture (Molnar, 2023), the more complex DNNs are difficult to interpret (Arrieta et al., 2020). This may lead to unexpected learning behavior by the model (e.g. learning spurious correlations

of features rather than the features themselves), which, for example, could result in the model's training set performance not translating to unseen data (e.g. Feldman & Zhang (2020); Xiao et al. (2020)).

As a result, interpretability of DNNs is an active field of research where there is an increasing focus on ensuring that DNNs make the *right decisions for the right reasons* (Ross et al., 2017). Popular post-hoc model evaluation techniques Ribeiro et al. (2016); Selvaraju et al. (2017) have been proposed to interpret model decisions, but these approaches usually require model retraining to correct unwanted model behavior. Therefore, a different approach involves enforcing priors on the feature attributions of DNNs during training, thereby preventing the network from learning spurious correlations in the first place. With these priors, the feature attributions of a model are guided to emphasize key regions within the input that are most influential in the model's decision-making process while also providing localized explanations that are easily interpretable by the user. Thus, model guidance using attribution priors is an approach to make DNNs more interpretable by jointly optimizing on the primary task, e.g. classification, as well as on the localization of attributions (e.g. Ross et al. (2017), Shen et al. (2021)). Previous studies, however, have not evaluated how to effectively train using attribution priors for model guidance as model guidance has been studied only using a limited set of attribution methods, localization losses and models under different evaluation settings. Moreover, as segmentation masks are required for guiding models during training, the annotation cost poses an issue for practical adaption. The study by Rao et al. (2023) addresses these issues by conducting an in-depth analysis on how to:

- *effectively* guide models using attribution priors on large, real-world datasets by evaluating the design elements along the directions of attribution methods, localization losses and model architectures.

- *efficiently* guide models using bounding boxes instead of segmentation masks and evaluate robustness of guidance techniques under limited or overly coarse annotations.

## 1.1 Scope of Reproducibility

In the following section, the main claims made by the authors are presented. The labels **R1**-**R9** of the claims follow the ones in the original paper (Rao et al., 2023) and will be used throughout this report to identify the corresponding findings:

- **Comparing loss functions for model guidance**: The Energy loss yields the best trade-off between accuracy and the EPG score (**R1**), whereas the $L_1$ loss provides the best trade-off between accuracy and the IoU score (**R2**). Furthermore, the Energy loss focuses best on on-object features by suppressing background features withing the bounding box (**R3**).

- **Comparing models and attribution methods**: The B-cos attribution method provides the best baseline performance (IoU, EPG) as well as the highest gain in EPG and IoU when trained with model guidance (**R4**). Additionally, all models show increased performance in IoU and EPG score when the model guidance is applied at the final layer (**R5**).

- **Improving model accuracy with model guidance**: While some models (Vanilla, $\mathcal{X}$-DNN) yield an increase in both the localization metric *and* the accuracy score (**R6**), the vast majority of models showcases a trade-off between localization and accuracy.

- **Efficiency and robustness considerations**: The localization performance (EPG or IoU) can be significantly improved over the baseline models even with a very limited number of annotated training samples (**R7**). In contrast to the $L_1$ loss, the Energy loss exhibits a high level of robustness against annotation errors, consistently yielding reliable results even when trained on samples with dilated bounding boxes (**R8**). Applying model guidance at intermediate or at the final layer leads to a speed-up in training while maintaining similar input-level attributions as input-level guidance (**R9**).

- **Effectiveness against spurious correlations**: Finally, the authors evaluate model guidance on a synthetically constructed dataset and claim that even a small number of annotations (1%) improves the model's generalization under distribution shifts at test time.

Our reproducibility study will aim to reproduce the claims **R1**-**R8**. Claim **R9** was outside of the main scope of the paper and would have come with great computational complexity, requiring significant resources to fully evaluate and verify, and is therefore not reproduced by us. The rest of this paper is organized as follows: Section 2 provides background information on model guidance and insight into technical concepts necessary for understanding the remainder of the paper. Section 3 summarizes the models, datasets, and the experimental setup we employed for reproducing the findings of the original paper. Section 4 presents the results of the reproduced experiments and introduces the findings from our additional investigations. Finally, Section 5 summarizes the work conducted in this paper and assesses the level of difficulty of reproducing the original paper.

## 2 Model guidance using attribution priors

In the original paper, the authors evaluate different combinations of attribution methods, localization losses, and evaluation metrics, for model guidance. We will shortly give a background on each of these components.

### 2.1 Attribution methods

To guide the model using attribution priors, three differentiable attribution methods (IxG (Shrikumar et al., 2017), GradCAM (Selvaraju et al., 2017), and IntGrad (Sundararajan et al., 2017)) with good localization properties were evaluated. Additionally, B-cos attributions (Böhle et al., 2022) from the inherently interpretable B-cos models were assessed.

**IxG** (Shrikumar et al., 2017): To compute the attribution for a model at the $i^{th}$ layer, IxG which computes the element-wise product $\odot$ of the input of the $i^{th}$ layer and the gradients of the $k$-th output w.r.t. the input to the layer, i.e. $X_i \odot \nabla_{X_i} f_k(X_i)$. Using this product, we can compute the linear contribution of a given input pixel to the model output for a piece-wise linear models such as DNNs with ReLU activations.

**GradCAM** (Selvaraju et al., 2017): At the $i^{th}$ layer, GradCAM computes the attribution as a ReLU-thresholded, gradient-weighted sum of the activations of a layer. It is calculated as $\text{ReLU}\left(\sum_c \alpha_{i,c}^k \odot U_i\right)$, where $c$ denotes the channel dimension, $\alpha_{i,c}^k$ denotes the average-pooled gradients of the $k$-th output w.r.t. the input to the layer $i$ and $U_i$ denotes the activations to the layer $i$.

**IntGrad** (Sundararajan et al., 2017): To calculate the attributions, IntGrad takes an axiomatic approach proposed by (Sundararajan et al., 2017) and formulates as the integral of gradients over a straight line path from a baseline reference input to the given input X. However, calculation of this integral requires several gradient computations, making it computationally expensive for use in model guidance. As a result, the authors use $\mathcal{X}$-DNN models that allow for an exact computation of IntGrad in a single backward pass, which is implemented equivalently to the calculation of attributions using IxG (Hesse et al., 2021).

**B-cos** (Böhle et al., 2022): B-cos attributions are calculated by using the inherently explainable B-cos networks, which encourage the alignment between the input $\mathbf{x}$ and the dynamic weight matrix $\mathbf{W(x)}$ during optimization. The attributions are then generated by the element-wise product of the dynamic weight matrix with the input, i.e. $\mathbf{W(x)} \odot \mathbf{x}$, which represents the contribution of the input pixels to the class $k$.

### 2.2 Evaluation metrics

The performance of the models in the classification task was evaluated using the F1 score. For the assessment of performance in the localization task, the two metrics listed below were employed.

**IoU metric** (Gao et al., 2022b): The IoU measures the intersection of the ground truth annotation mask with the binarized attribution maps and normalizes it by their union.

**EPG metric** (Wang et al., 2020): The EPG metric measures the fraction of positive attributions that fall within the attribution mask (see Sec. 3.5.1 and Fig. 1 for more details). Unlike IoU, EPG does not binarize the attribution maps and thus keeps information about the relative importance of each input region.

### 2.3 Localization losses

To effectively localize the attributions in an image, different localization losses were evaluated.

**$L_1$ loss** (Gao et al. (2022b); Gao et al. (2022a)): The $L_1$ loss minimized the distance between the annotation masks and the normalized positive attributions $\hat{A}_k^+$, thereby guiding the model attributions towards uniform attributions within the mask and suppressing attributions outside the mask.

**Per-pixel Cross-Entropy (PPCE) loss** (Shen et al., 2021): The PPCE loss applies a binary cross entropy loss between the mask and the positive attributions $\hat{A}_k^+$, thus maximizing the attributions within the mask. However, there are no constraints on the attributions outside the mask, thereby resulting in potential spurious correlations.

**RRR\* loss** (Ross et al., 2017): RRR\* loss regularizes attributions outside the ground truth masks, thereby constraining attributions to lie within the mask. While it does not introduce a uniformity prior similar to the $L_1$ loss, it also does not explicitly promote high importance attributions inside the masks.

**Energy loss**: The authors propose the use of the negated EPG metric (Wang et al., 2020) as a loss function, as it is fully differentiable. They claim that the energy loss jointly optimizes for high attribution energy within the mask and low attribution energy outside the mask.

## 3 Methodology

In this section, we introduce details about the models, datasets, and the experimental setup (depicted in Figure 11) that we followed in our attempt to reproduce the work by Rao et al. (2023). The majority of the code required for conducting our experiments was publicly accessible through the authors' repository[1]. However, the code for running the experiments to test effectiveness against spurious correlations was not present and we had to re-implement it based on the descriptions that were given in the paper. Furthermore, we run several additional experiments to improve evaluation, re-assess the feasibility of using segmentation masks given the findings in the paper, as well as to discuss the impact of context that was not considered by the original authors.

### 3.1 Model descriptions

All models are pretrained on ImageNet (Russakovsky et al., 2015) using a ResNet-50 (He et al., 2015) backbone and fine-tuned on the target dataset with model guidance. The architecture varies based on the attribution method: vanilla ResNet-50 for IxG and GradCAM, $\mathcal{X}$-DNN ResNet-50 for IntGrad, and B-cos ResNet-50 for B-cos attributions. Optimization of attributions at different layers and various localization losses, detailed in Section 2.3, were explored. Table 1 provides a concise summary, and more detailed information is available in the Appendix (Table 6).

### 3.2 Datasets

Both the baseline and the guided models are fine-tuned on the target datasets: **PASCAL VOC 2007** (Everingham et al., 2009) and **MS COCO 2014** (Lin et al., 2014). Due to limited computational resources, we conduct the reproducibility experiments only on the COCO dataset, which is chosen for its diversity and complexity, providing a more comprehensive testing ground for object detection models than the VOC dataset (Wahab et al., 2022). Furthermore, experiments for testing effectiveness against spurious correlations utilize the **Waterbirds-100** dataset (Sagawa et al. (2019); Petryk et al. (2022)) – a synthetic dataset featuring different bird types (waterbirds vs. landbirds) on different background types (water vs. land). The challenge of this dataset is that waterbirds and landbirds are perfectly correlated with water and land backgrounds, respectively, on the train and validation sets, but are equally likely to occur on both background types in the test set. Table 5 in the Appendix provides further insight into the datasets.

---

[1]https://github.com/sukrutrao/Model-Guidance

Table 1: **Model configurations used by Rao et al. (2023).** Additional configurations tested by the original authors that we did not attempt to reproduce due to limited computational resources are shown in light grey. Each column stacks multiplicatively, resulting in a total of 18 baseline models out of which we reproduced 3, and 480 guided models, out of which we reproduced 18.

| BASELINE MODELS | | | | | |
|---|---|---|---|---|---|
| **Backbone** | **Dataset (#Epochs)** | **Batch size** | **Learning rate** | **Attribution layer** | **Localization loss ($\lambda_{\text{loc}}$)** |
| Vanilla, $\mathcal{X}$-DNN B-cos | COCO (60), VOC (300) | 64 | $10^{-4}, 10^{-3}, 10^{-5}$ | - | - |
| GUIDED MODELS | | | | | |
| **Backbone (Attribution method)** | **Dataset** | **Batch size** | **Learning rate** | **Attribution layer** | **Localization loss ($\lambda_{\text{loc}}$)** |
| Vanilla (IxG, GradCam), $\mathcal{X}$-DNN (IntGrad), B-cos (B-cos) | COCO (10), VOC (50) | 64 | $10^{-4}$ | Input, Final, Mid1, Mid2, Mid3 | $L_1$ ($5{\times}10^{-3}, 1{\times}10^{-3}, 1{\times}10^{-2}$) Energy ($1{\times}10^{-3}, 5{\times}10^{-4}, 5{\times}10^{-3}$) RRR* ($1{\times}10^{-5}, 5{\times}10^{-6}, 5{\times}10^{-5}$) PPCE ($5{\times}10^{-4}, 1{\times}10^{-4}, 1{\times}10^{-3}$) |

## 3.3 Hyperparameters

In reproducing the experiments, we used the authors' specified hyperparameters whenever available. For missing values, we estimated them based on informed judgments. For instance, the missing batch size for training baseline models was set to match guided models. In cases where the authors tested multiple values without specifying the best-performing one, we set the hyperparameter to the median value due to limited computational resources for testing all configurations.

## 3.4 Implementation issues

The author's implementation of IoU metric requires binarised attribution maps, for which the threshold value for IoU was obtained from a heldout subset of the validation data. Neither the optimised threshold values nor the heldout validation subset were made available in the authors' original code. Moreover, training of guided models on the PPCE loss resulted in NaN loss during training, even though it required executing only a single command with appropriate parameters specified by the authors. Therefore, we were not able to reproduce the results requiring IoU metrics and PPCE loss, which are, thus, omitted from evaluation.

## 3.5 Extensions of the original paper

### 3.5.1 X-SegEPG

The original EPG metric (Wang et al., 2020) has a notable limitation in treating all attributions within the bounding box uniformly. This approach can lead to high scores even if the model is not providing accurate explanations, as it may focus on surrounding context within the imprecise bounding box. Rao et al. (2023) recognized this issue and introduced Segmentation EPG (SegEPG) – a metric that refines the evaluation process by utilizing segmentation masks, counting only those attributions falling within these masks as correct. Attributions located within the bounding box but outside the segmentation mask are considered as incorrect. However, we argue that SegEPG has its own limitations, as it disregards attributions outside the bounding box, allowing scenarios where high scores may not reflect genuine accuracy.

To address this, we propose Extended Segmentation EPG (X-SegEPG) (Eq. 1), which combines aspects of both the classic EPG and SegEPG (see Appendix F for a formal comparison of the metrics and detailed notation). Just like SegEPG, X-SegEPG considers only attributions on the segmentation mask as correct. However, instead of considering only attributions outside the segmentation mask, but within the bounding box as incorrect, we suggest to count all attributions outside the segmentation mask as incorrect, regardless of whether they fall within the bounding box or not. Formally, we define X-SegEPG for class $k$, as:

$$\text{X-SegEPG}_k = \frac{\sum_{h=1}^{H}\sum_{w=1}^{W} S_{k,hw} A_{k,hw}^{+}}{\sum_{h=1}^{H}\sum_{w=1}^{W} A_{k,hw}^{+}} \tag{1}$$

where $H$ and $W$ are the height and width of the image, $S$ is the segmentation mask, and $A^{+}$ is the positive attribution mask F. Figure 1 illustrates the common issues with the standard EPG metric (Wang et al., 2020) and SegEPG extension proposed by Rao et al. (2023). The standard EPG metric will yield a perfect score of 1.0, as long as all attributions are located within the bounding box. However, in the example shown in the figure, none of the attributions are on the object itself, meaning that the model was still not *right for the right reasons*. SegEPG no longer counts attributions outside of the segmentation mask as correct, but there is still a potential source of error. As shown in the figure, SegEPG returns a perfect score, as long as all attributions within the bounding box are exactly on the object, even though the model might still be largely focusing on spurious information outside the bounding box. X-SegEPG proposed in our work eliminates the two error types occurring in previous metrics, guaranteeing it does not score perfectly in scenarios where errors are present. This modification ensures a more accurate assessment of model performance.

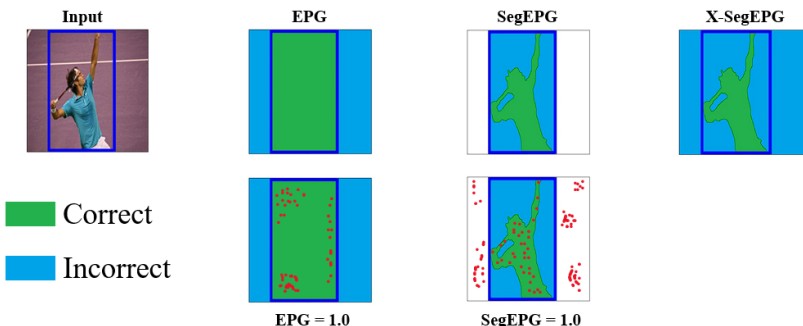

Figure 1: **Visual comparison of the different EPG variants and common error sources.** Red dots represent positive attributions, which are stored as an attribution mask indicating the contribution of each pixel to the output. Green and blue areas indicate which attributions are considered correct and incorrect, respectively, under the given metric.

### 3.5.2 EPG vs. Bounding box size

Another limitation of the EPG metric stems from the fact that it is easier to achieve high EPG scores when the bounding box is large. As we increase the bounding box, the EPG is bound to increase by definition. In cases of the bounding box covering most of the image (e.g. a close-up image of the target object), the majority of attributions are likely to fall within the bounding box, primarily due to its extensive size. This inherent characteristic of larger bounding boxes makes achieving high EPG scores comparatively easier, making the results often too optimistic. In contrast, a smaller bounding box introduces a much greater margin for error, making it less likely for attributions to be accurately confined within it. This discrepancy illustrates a bias within the EPG metric, which tends to favor larger bounding boxes over smaller ones and we propose a way to tackle this issue, in order to achieve more realistic values.

### 3.5.3 The impact of context in image classification

Throughout the original paper, the authors worked with the premise of model guidance that the relevant pieces of information for image classification should be primarily focused on the object of interest. This is particularly true for out-of-distribution (OOD) scenarios where contextual information might be misleading (e.g. a waterbird on land). However, the information found within the context is often not spurious for in-distribution images. Previous research has investigated this phenomenon as well (e.g. Divvala et al. (2009), Wang & Zhu (2023)) and showed that context can enhance object classification and detection tasks (Oreski, 2023).

We further hypothesized that there are scenarios where context is, in fact, necessary for accurate and confident classification of the target object, at least for humans. To test this hypothesis and understand the role of context in human perception, we conducted a study with human participants. By understanding how humans leverage context in certain situations, we can design models that mimic these strategies, potentially leading to more robust and accurate performance. If humans can decide when to use context to aid their decision or when to focus only on the on-object features, why should we constrain a model from doing so?

For this purpose, we selected a set of 6 images exemplifying such cases and show an example in Figure 8a (the remaining images are found in Appendix J). We then conducted a survey in which participants were initially shown only the cropped version of the image, stripped of the context, and asked to categorize the object into one of the given categories. Subsequently, we presented them with the full image, inquiring if their decision changed and, if so, to explain the reason for the change.

### 3.5.4 Segmentation masks and sparse annotations

Finally, we evaluate the robustness of the energy loss given the findings in the original paper. In the introduction, the authors argued that segmentation masks are extremely expensive to obtain for large datasets and show that much cheaper bounding boxes also lead to promising results. They later show that even when using only a fraction of annotations during training, they manage to achieve notable results (**R7**).

This opens the door for reassessing the feasibility of using segmentation masks – if we can achieve comparable performance with a very limited number of annotations, it might become affordable to use segmentation masks, which provide the ground truth, instead of the less precise bounding boxes. We explore this on the PASCAL VOC 2007 dataset and using the best performing B-cos model with input layer guidance. We retrain the model with 8% of data annotated with segmentation masks and compare it to using 8% of bounding box annotations, as well as the fully annotated dataset, to check if there is a significant improvement in performance.

## 3.6 Computational requirements

We conducted our experiments using an Nvidia HGX A100 Tensor Core GPU. Given limited computational resources, the reported experiments were run once and are thus to be taken indicative and not conclusive. The total computational expense for this reproducibility study amounts to 74.75 GPU hours, which corresponds to an estimated emission of 7.89 kgCO$_2$eq.[2]

# 4 Results

## 4.1 Results reproducing original paper

**Comparing loss functions for model guidance:** The authors' first claim **R1** was reproducible by us, as is shown in Figure 4.1. The models trained using Energy loss outperformed models trained using the other loss functions in all model configurations and consistently beat the baselines in terms of localisation performance based on EPG score. We further managed to reproduce the trends that led to the authors third claim **R3**. The Energy loss outperforms the L1-loss with respect to its ability to localize attributions to the object as opposed to background regions within the bounding boxes (see Appendix K for details).

**Comparing models and attribution methods:** Claim **R4** is again confirmed by our findings. When comparing the sub-plots on the top row of Figure 4.1, the B-cos model on the right shows the highest baseline performance for classification as well as localisation, in addition to achieving the largest increase in performance after model guidance. Unlike the other models, this increase in performance post-guidance is not limited to localisation (when using $L_1$ or Energy loss) but also manifests itself in an increased classification performance. This suggests that the model-inherent B-cos attributions more closely align with the intrinsic features of the dataset, but are also more receptive to guidance and can thus not only aid in better explaining model behaviour, but improve performance too.

---

[2]Estimations were conducted using the MachineLearning Impact calculator presented in Lacoste et al. (2019).

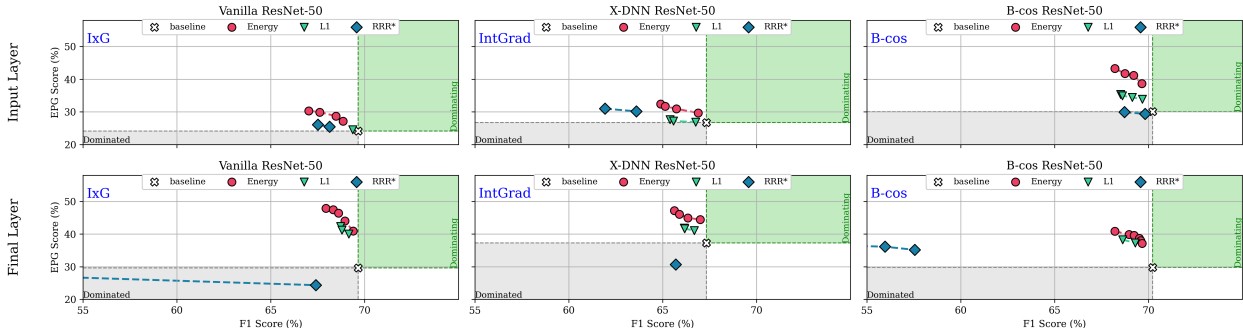

Figure 2: **EPG vs F1** on COCO for different localisation loss functions (markers) used during training, models (columns) and model guidance layers (rows).

The author's fifth claim **R5** could not be fully substantiated by our findings in Figure 4.1, as our results show this only taking place for models trained using either $L_1$ or Energy loss. With EPG as localisation metric, models trained with RRR* loss are either dominated by the baseline performance or can only beat the baseline in terms of localisation performance by sacrificing classification performance. This warrants further investigation into loss function-specific effects on localisation and classification outcomes.

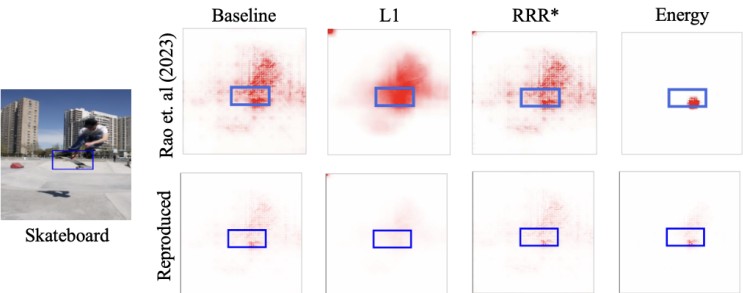

Figure 3: **Comparison of qualitative results of Rao et al. (2023) and our reproduced results.** While the magnitudes of the attributions in our results differ from the original ones, we do observe the same trends.

**Improving model accuracy with model guidance:** We achieved slightly different results regarding **R6**. Our findings generally show this increase in both performance metrics instead for B-cos models, for which most configurations dominate the respective baseline. The overall trend when considering all possible configurations that classification and localisation experience a trade-off relationship however is supported by our results, most commonly showing an increased localisation performance for the price of slightly worse classification accuracy.

**Efficiency and robustness considerations:** Claim **R7** could indeed be confirmed by our experiments, although the trend is not as clear as in the original paper. As Figure 14 shows, we do see that only annotating a fraction of the training data still results in localisation performance gains on the COCO dataset. We noticed however that when annotating 10% of training images, the models showed a larger increase in localisation performance at the expense of a larger decrease in classification performance than the trend would suggest. This slight deviation in behaviour at different annotation percentages might be a consequence of the different dataset being evaluated, as the original paper only evaluates the sparse annotation performance on the VOC dataset, in both the main paper and supplement.

**R8** was strongly supported by our findings. Figure 5 depicts how even with bounding box annotations that are up to 50% larger than ground-truth the models trained with Energy loss can exhibit consistent increases in localisation performance. Models trained using $L_1$ loss however start to get dominated by the baseline in terms of both localisation and classification performance at 25% larger annotations already. This

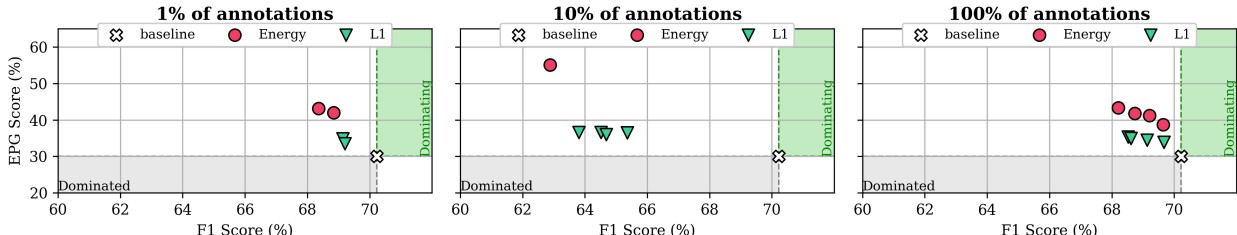

Figure 4: **EPG results with limited annotations** on COCO for a B-cos model guided at the input layer, optimised with Energy or $L_1$ loss (markers).

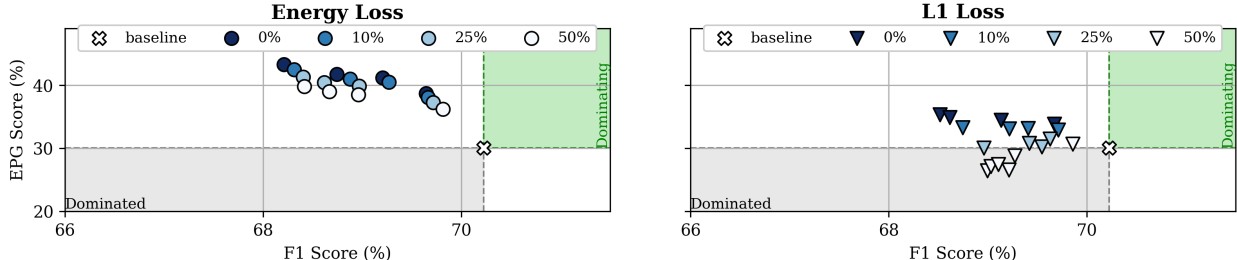

Figure 5: **Quantitative results for dilated bounding boxes** on COCO for a B-cos model guided at the input layer. Localisation performance based on EPG score is plotted against F1 score for models trained with increasingly erroneous annotations (larger bounding boxes). Models trained with Energy loss yield consistent results despite training with heavily dilated bounding boxes (left), whereas the results of the $L_1$ loss (right) worsen markedly.

demonstrates the suitability of Energy loss as a loss function for training in these bounding box settings, as it is able to more finely allow the model to place attributions on the object in question even within large bounding boxes.

**Effectiveness against spurious correlations:** Model guidance on the Waterbirds-100 dataset was performed under two settings: (1) distinguishing between waterbirds and landbirds using the region within the bounding box as mask (**conventional setting**) and (2) classifying the background (water or land) using the region ouside of the bounding box as mask (**reversed setting**). Table 2 compares our results with those from the original paper. Our results confirm the claims of the authors – the guided models improve the performance compared to the baseline and training on energy loss leads to better performance across all configurations (shown in bold). Qualitative results are found in the Appendix 16.

| Model | Conventional | | | | Reversed | | | |
| | Worst | | Overall | | Worst | | Overall | |
| | Orig. | Repr. | Orig. | Repr. | Orig. | Repr. | Orig. | Repr. |
|---|---|---|---|---|---|---|---|---|
| Baseline | 43.4 | 40.5 | 68.7 | 65.4 | 56.6 | 59.5 | 80.1 | 83.3 |
| Energy | **56.1** | **55.0** | **71.2** | **72.4** | **62.8** | **63.1** | **83.6** | **84.8** |
| $L_1$ | 51.1 | 52.1 | 69.5 | 70.1 | 58.8 | 57.2 | 82.2 | 83.7 |

Table 2: **Comparison of original and reproduced results on Waterbirds-100.** All of the results are very close to those from the original paper. We only run the models once, due to computational limitations, whereas the authors report average scores across four different runs, which explains the slight differences.

## 4.2   Results beyond original paper

**X-SegEPG:** In Section 3.5.1, we theoretically justify the robustness of our proposed metric. Here, we also provide quantitative insight by evaluating the top-performing B-cos model with input layer guidance on 100 images from the COCO test dataset. We focused on images with a single class and label for easier interpretation. As shown in Table 3, the average X-SegEPG is significantly lower than EPG and SegEPG, showcasing that the latter metrics can yield overly optimistic values.

| EPG | | SegEPG | | X-SegEPG | |
|---|---|---|---|---|---|
| Val. | Std. | Val. | Std. | Val. | Std. |
| 0.35 | $\pm 0.23$ | 0.33 | $\pm 0.25$ | 0.16 | $\pm 0.2$ |

Table 3: **Comparison of EPG, SegEPG, and X-SegEPG.** Average metric values and standard deviations for 100 single-class, single-label COCO test images. X-SegEPG provides a more realistic evaluation than overly optimistic EPG and SegEPG.

**EPG vs. Bounding box size:** We find a strong positive correlation between the size of the bounding boxes and the EPG score ($r = 0.8209$, $p < 0.001$ based on Pearson's correlation coefficient), indicating that as the size of the bounding boxes increases, the EPG score also tends to increase. The analysis was conducted with a B-cos model which was fine-tuned with attributions at the input layer on the COCO dataset. A graphical representation of the correlation is provided in Appendix G (Figure 15). The correlation leads to a bias that unfairly favors larger bounding boxes and hence larger objects. This is problematic because it introduces misleading performance assessments, as models may appear to perform better simply due to the increased number of pixels in larger bounding boxes. This bias can mask true model deficiencies and reduce generalizability, making models less robust across different object sizes.

With a simple normalization of the EPG score by the square root of the total number of pixels covered by one or more bounding boxes of the same class, we were able to decrease the correlation ($r = 0.1680$, $p < 0.001$). Therefore we propose a modified version of the Energy loss, coined Energy$^*$ loss, which accounts for the different bounding box sizes. This loss is proposed in Eq. 2 as:

$$\mathcal{L}_{\text{Energy}^*} = 1 - \frac{1}{(\text{BB Area})^\alpha} \times \text{EPG} \tag{2}$$

where we set the hyperparameter $\alpha = 0.5$. We trained the B-cos model with input layer guidance on the Energy loss and the proposed Energy$^*$ loss on the smaller VOC 2007 dataset. From Figure 6, we observe that B-cos model trained on Energy$^*$ loss have a higher F1-score but lower EPG compared to model trained on Energy loss.

**The impact of context in image classification:** We present an example survey image depicting a real car in Figure 8a. Furthermore, this section points out to the most important results that support our original hypothesis. We also conducted appropriate statistical tests that show significance of our findings. The full details, including the remaining images, survey results and statistical analysis are presented in Appendix J. From Figures 8b and 8c we can see a significant increase in accuracy of participants answers once they incorporated contextual information. In total, there were 99 occasions when participants changed their decision after seeing the full image and out of that 80% were changes from an incorrect to a correct classification (Figure 10). Furthermore, Figure 9 shows that there was a consistent improvement in classification accuracy after seeing the entire image. In addition, when being asked what predominantly guided their decision – the on-object or contextual information – participants chose the former option only 24% of the times.

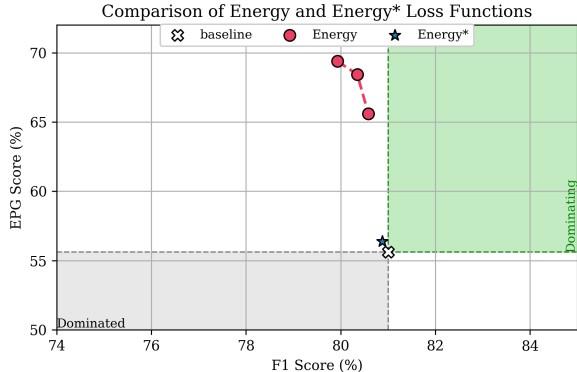

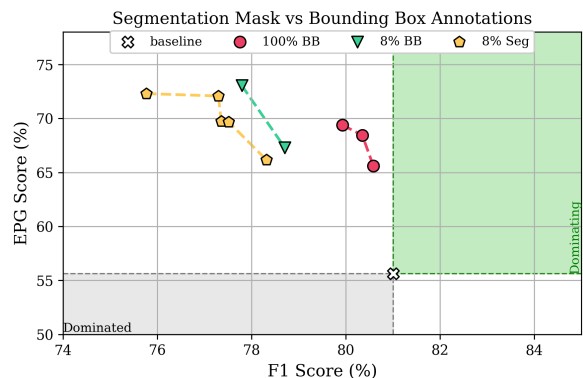

Figure 6: **Evaluation of Energy\* loss vs Energy loss**. The Energy\* loss outperforms Energy loss on the F1-score but not on EPG.

Figure 7: **Evaluation of robustness of energy loss with sparse annotated segmentation masks**. The energy loss focuses on on-object features even within the bounding box and as a result, use of sparse segmentation mask for model guidance does not improve the performance.

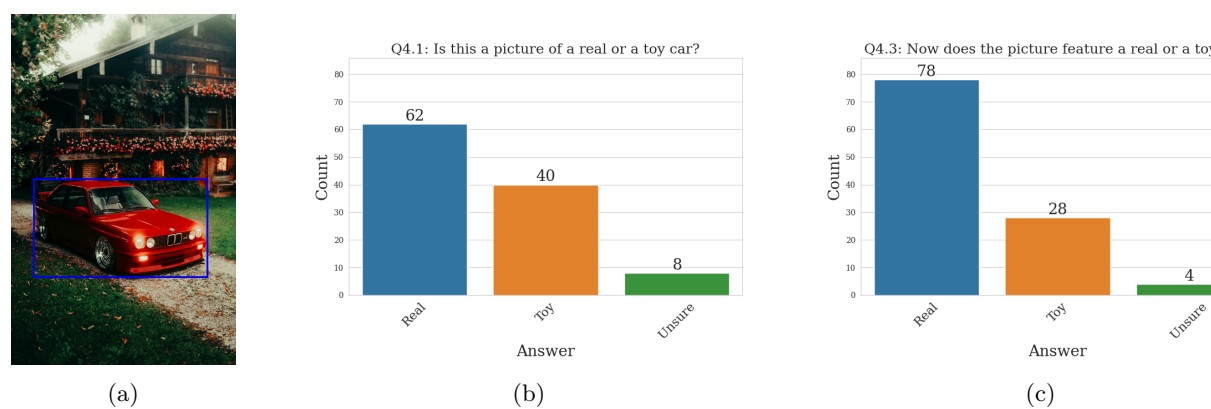

Figure 8: **Example survey question and answer distribution.** (a) Participants were first shown only the part of the image within the shown bounding box and the full image only afterwards. (b) Answers when only shown the cropped version. (c) Answers after seeing the full image.

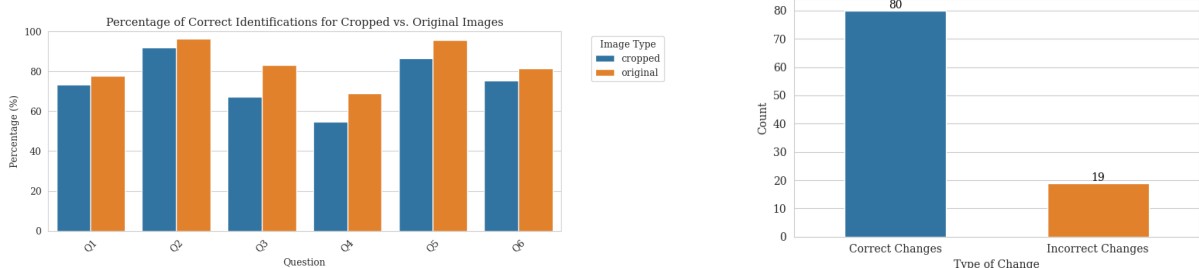

Figure 9: **Correct identification with and without context.**

Figure 10: **Decision changes after observing context.**

**Segmentation masks and sparse annotations:** To analyze the effect of the using sparse segmentation masks, we train B-cos models with input layer guidance and compare results with 8% and 100% bounding box annotated VOC dataset. From figure 7, we can observe that the 8% sparsely and 100% annotated BB

models perform better than the 8% segmentation mask annotated model. Interestingly, this finding indicates that it is not just sufficient, but preferable to use bounding boxes. This indicates that even the local context, an inherent part of bounding boxes, might be beneficial for boosting model performance, further supporting our hypothesis about the importance of contextual cues in certain scenarios.

## 5 Discussion

### 5.1 Reproducibility assessment

In this work, we aimed at reproducing the main claims of the paper by Rao et al. (2023). The paper presented ten main claims distributed over a total of five experiments out of which we were able to test eight. Altogether, we conclude that the majority of the claims are reproducible, with some minor fluctuations. Table 4 gives a brief reproducibility summary. Finally, we give our assessment of the overall workflow:

**What was easy:** The original paper provided most of the details for the experimental settings and hyper-paremeters. Furthermore, most of the original implementation was publicly accessible, as well as scripts for downloading and pre-processing the VOC and COCO dataset and a description on how to train baseline and guided models.

**What was difficult:** Even though most of the code was present, it lacked proper documentation, so it took some effort to fully understand it. Furthermore, we had to implement certain missing parts of the code and make educated assumptions where the information was lacking.

**Communication with original authors:** We contacted the original authors to seek clarification on such aspects of the experimental setup and get access to missing code. We received a relatively late response from the authors and our questions were only partially answered, so we had to adhere to our choices.

Table 4: **Reproducibility analysis summary.** Experiments that were not attempted are marked as N/A.

| Experiment | Claim | Reproducible? | Notes |
|---|---|---|---|
| Comparing loss functions for model guidance | **R1** | Yes | - |
|  | **R2** | No | Issue with IoU |
|  | **R3** | Yes | - |
| Comparing models and attribution methods | **R4** | Yes | - |
|  | **R5** | Partially | Does not hold for RRR* loss |
| Improving model accuracy with model guidance | **R6** | Partially | Classification-localisation trade-off, except for B-cos models |
| Efficiency and robustness considerations | **R7** | Yes | Same trends, different magnitudes |
|  | **R8** | Yes | - |
|  | **R9** | N/A | - |
| Effectiveness against spurious correlations | Sparse annotations improve model generalization | Yes | - |

### 5.2 Extended work and future research directions

In addition to our efforts to reproduce the original results by Rao et al. (2023), we proposed several extensions to the original work. This involved making the evaluation more robust through EPG and Energy loss modifications. We also re-assessed the importance of context for model guidance and the feasibility of using segmentation masks given the findings in the original work.

We suggested making an extension to the EPG metric and showed why it was more robust than the one the author's proposed. We also proposed adding a penalty term to the Energy loss function, to account for the bounding box size. However, to maximize robustness, further experiments are required to determine the appropriate values of $\lambda_{loc}$ and $\alpha$ hyper-parameters, alongside longer training time for optimization. In addition, it might be worth further analyzing segmentation literature for potentially more suitable attribution losses invariant to bounding box size (e.g. Dice loss (Milletari et al., 2016)).

Furthermore, we re-evaluated the feasibility of leveraging segmentation masks, given the authors' findings (**R7**). We find that using a small portion of segmentation mask annotations (8%) yields a comparable performance to using the same proportion of bounding boxes. This further confirms the authors' claims that the Energy loss provide on-object attention even when using bounding boxes, rendering segmentation masks redundant.

Moreover, our survey results signal that there are certain scenarios where context information plays a crucial role for accurate object detection. Keeping in mind that our study involved a small number of images and participants, thus raising potential biases, we hope that it still offers valuable insights. We believe it provides sufficient support for further investigation into the conditions under which contextual cues play a significant role and how to properly incorporate these cues into model guidance.

Finally, we acknowledge that in order to ensure the generalizability of our proposed extensions, they would benefit from being tested on a wider range of of models, datasets, and domains.

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

## A    Experimental Setup

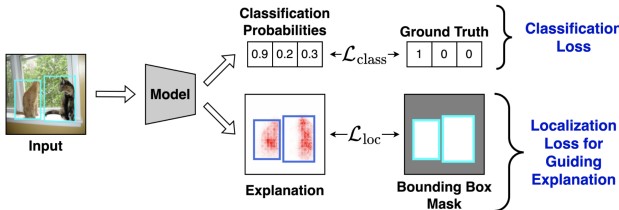

Figure 11: **Experimental setup of the study:** The experimental setup optimizes for both classification ($\mathcal{L}_{class}$) as well as the localization of the attributions within the human-annotated bounding boxes ($\mathcal{L}_{loc}$), thereby, guiding the model to focus on on-object features.

## B    Dataset details

Table 5 provides detailed information about the datasets. As a pre-processing step, all images were resized to 224×224 pixels to ensure consistent input dimensions to the network.

Table 5: **Datasets overview.**

| Task | Datasets | Number of Samples | | | | Classes | Description | URL |
|---|---|---|---|---|---|---|---|---|
| | | **Train** | **Val** | **Test** | **Total** | | | |
| Experiments **R1–R9** | PASCAL VOC 2007 | 2,501 | 2,510 | 4,952 | 9963 | 20 | - | Link |
| | MS COCO 2014 | 73,872 | 8,209 | 40,504 | 122,585 | 80 | - | Link |
| Effectiveness against spurious correlations | Waterbirds-100 | 4,795 | 1,199 | 5,794 | 11,788 | 2 | Experiments conducted under two settings: (1) conventional and (2) reversed | Link |

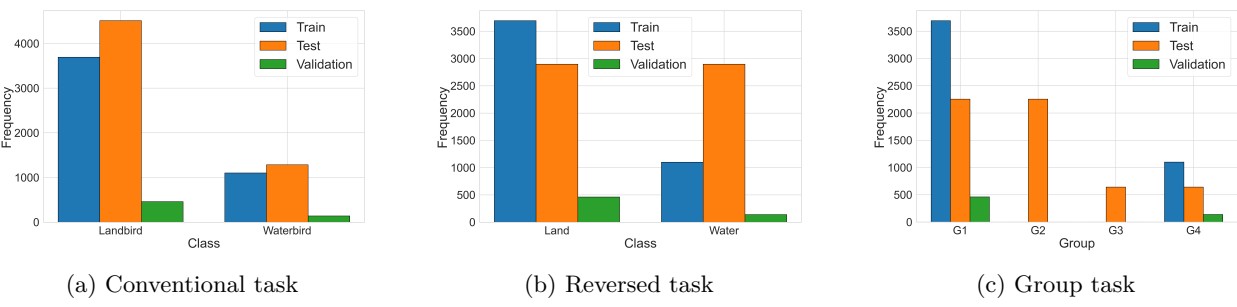

(a) Conventional task                    (b) Reversed task                    (c) Group task

Figure 12: **Label distribution for the Waterbirds-100 dataset across different settings**

In addition, Figure 12 provides label distribution for the Waterbirds-100 dataset across different settings. In the conventional setting (12a) there is significant class imbalance across all splits. In the reversed setting (12b), this is only the case for training and validation splits. The class imbalance was handled during training by the use of a weighted loss function. The weights were calculated manually based on the number of samples. Figure 12c provides a more fine-grained insight into the label distribution by showing per-group statistics, where the groups are: *Landbird* on *Land* (**G1**), *Landbird* on *Water* (**G2**), *Waterbird* on *Land* (**G3**) and *Waterbird* on *Water* (**G4**). As can be seen from the figure, only **G1** and **G4** were present during training, while all groups were present at test time. In the original dataset, the validation set also contained all groups, but **G2** and **G3** were removed by the authors during pre-processing in order not to leak the test

set distribution. The validation set originally contained 1199 as indicated in 5, of which 600 remained after the pre-processing.

## C  Experiment details

Table 6: **Hyperparameters/ Datasets and model details for the models used by Rao et al. (2023)**. Additional configurations/ datasets tested by the original authors that we did not attempt to reproduce due to limited computational resources are in light gray. The columns stack multiplicatively.

| BASELINE MODELS | | | | | | |
|---|---|---|---|---|---|---|
| **Backbone** | **Dataset (#Epochs)** | **Batch size** | **Learning rate** | **Attribution layer** | **Localization loss ($\lambda_{\text{loc}}$)** | **Note** |
| Vanilla, $\mathcal{X}$-DNN B-cos | COCO (60), VOC (300), Waterbirds (350) | 64 | $10^{-4}, 10^{-3}, 10^{-5}$ | - | - | - |
| **GUIDED MODELS** | | | | | | |
| **Backbone (Attribution method)** | **Dataset** | **Batch size** | **Learning rate** | **Attribution layer** | **Localization loss ($\lambda_{\text{loc}}$)** | **Note** |
| Vanilla (IxG, GradCam), $\mathcal{X}$-DNN (IntGrad), B-cos (B-cos) | COCO (10), VOC (50) | 64 | $10^{-4}$ | Input, Final, Mid1, Mid2, Mid3 | $L_1$ ($5{\times}10^{-3}, 1{\times}10^{-3}, 1{\times}10^{-2}$) 
 Energy ($1{\times}10^{-3}, 5{\times}10^{-4}, 5{\times}10^{-3}$) 
 RRR* ($1{\times}10^{-5}, 5{\times}10^{-6}, 5{\times}10^{-5}$) 
 PPCE ($5{\times}10^{-4}, 1{\times}10^{-4}, 1{\times}10^{-3}$) | - |
| **GUIDED MODELS (DILATED BOUNDING BOXES)** | | | | | | **Dilation percentage** |
| **Backbone (Attribution method)** | **Dataset** | **Batch size** | **Learning rate** | **Attribution layer** | **Localization loss ($\lambda_{\text{loc}}$)** | |
| B-cos (B-cos) | COCO (10), VOC (50) | 64 | $10^{-4}$ | Input | $L_1$ ($5{\times}10^{-3}, 1{\times}10^{-3}, 1{\times}10^{-2}$) 
 Energy ($1{\times}10^{-3}, 5{\times}10^{-4}, 5{\times}10^{-3}$) | 0, 10, 25, 50 |
| **GUIDED MODELS (SPARSE ANNOTATIONS)** | | | | | | **Annotation percentage** |
| **Backbone (Attribution method)** | **Dataset** | **Batch size** | **Learning rate** | **Attribution layer** | **Localization loss ($\lambda_{\text{loc}}$)** | |
| B-cos (B-cos) | COCO (10), VOC (50) | 64 | $10^{-4}$ | Input | $L_1$ ($1.00, 0.01, 0.100$) 
 Energy ($0.50, 0.05, 0.100$) | 1, 10, 100 |
| **GUIDED MODELS (WATERBIRD)** | | | | | | **Annotation percentage** |
| **Backbone (Attribution method)** | **Dataset** | **Batch size** | **Learning rate** | **Attribution layer** | **Localization loss ($\lambda_{\text{loc}}$)** | |
| B-cos (B-cos) | Waterbirds (60) | 16 | $10^{-5}$ | Input | $L_1$ ($5{\times}10^{-2}$) 
 Energy ($5{\times}10^{-2}$) | 1 |
| **EXTENSION: ENERGY vs ENERGY* LOSS** | | | | | | **Annotation percentage** |
| **Backbone (Attribution method)** | **Dataset** | **Batch size** | **Learning rate** | **Attribution layer** | **Localization loss ($\lambda_{\text{loc}}$)** | |
| B-cos (B-cos) | VOC (50) | 16 | $10^{-4}$ | Input | Energy* ($0.5$) 
 Energy ($0.5$) | - |
| **EXTENSION: SEGMENTATION MASK AND SPARSE ANNOTATION** | | | | | | **Annotation percentage** |
| **Backbone (Attribution method)** | **Dataset** | **Batch size** | **Learning rate** | **Attribution layer** | **Localization loss ($\lambda_{\text{loc}}$)** | |
| B-cos (B-cos) | VOC (50) | 8 | $10^{-4}$ | Input | $L_1$ ($5{\times}10^{-2}$) 
 Energy ($5{\times}10^{-2}$) | 8, 100 |

# D    Qualitative results

Table 7: **Qualitative results** for the COCO dataset with different models, loss functions, and at different layers.

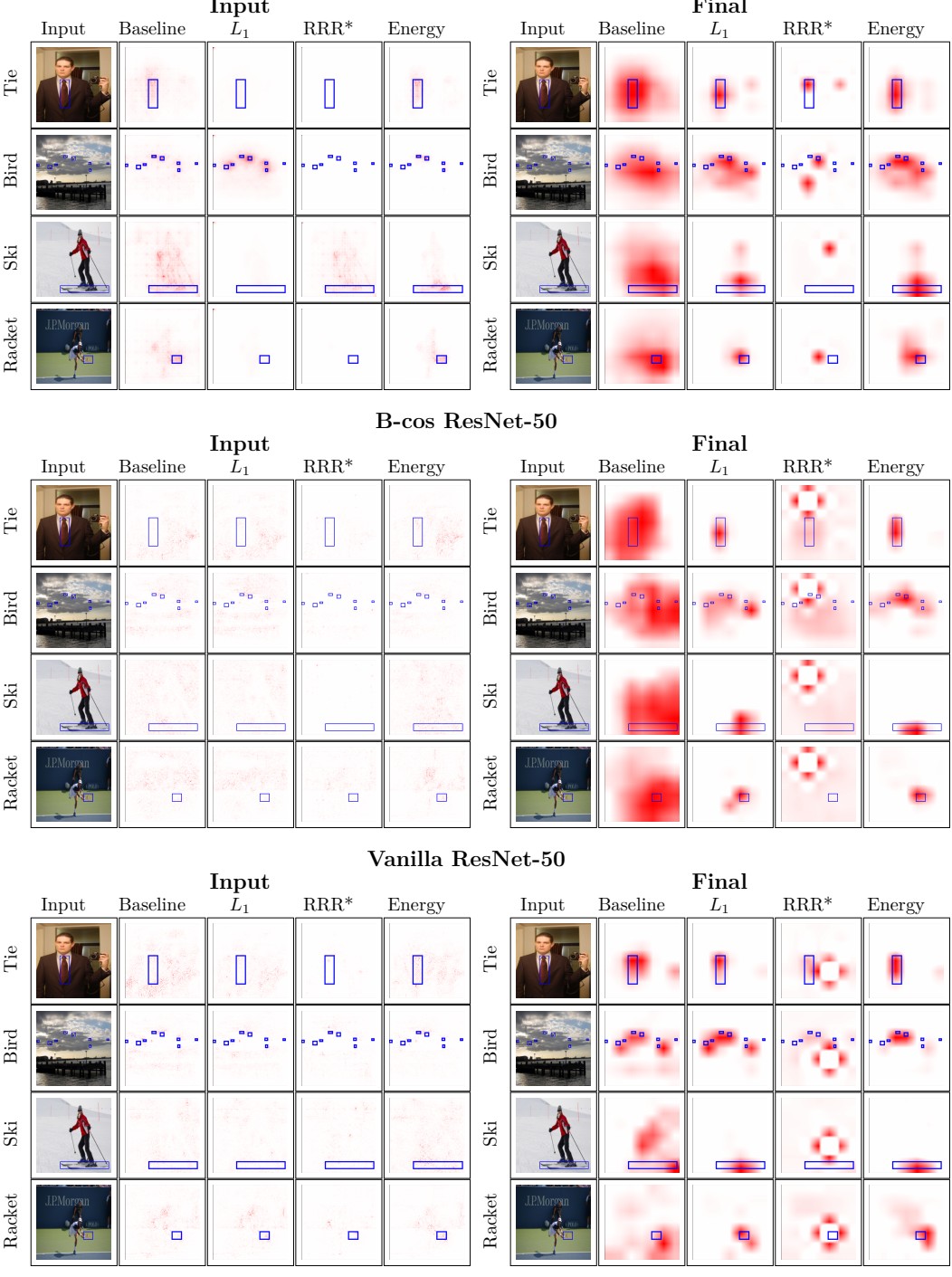

## E   Qulitative results for dilated bounding boxes

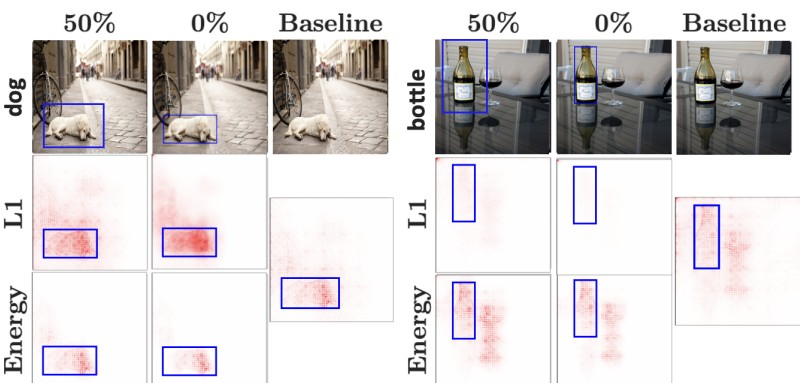

Figure 13: **Qualitative results for dilated bounding boxes** for a B-cos model at input. Examples for attributions (rows 2+3) of models trained with dilated bounding boxes (row 1).

## F   EPG metrics comparison

For a multi-label classification setting with $K$ classes, consider an input image $X \in R^{C \times H \times W}$ and one-hot encoded image labels $y \in \{0, 1\}^K$. With the attribution map for class $k$ for input image $X$ using a classifier $f$ denoted by $A_k \in \mathbb{R}^{H \times W}$, 1. $\hat{A}_k = \frac{A_k}{max(abs(A_k))}$ refers to the normalized attributions, and 2. $\hat{A}_k^+ = \frac{A_k^+}{max(A_k^+)}$ refers to the positive normalized attributions with $A_k^+$ denoting the positive component of the attributions. Finally, $M_k \in \{0, 1\}^{H \times W}$ represents the binary mask for class k, which is given by the union of bounding boxes of all occurrences of class $k$ in $X$.

| Metric name | Formula |
|---|---|
| $\text{EPG}_k$ | $\dfrac{\sum_{h=1}^{H} \sum_{w=1}^{W} M_{k,hw} A_{k,hw}^+}{\sum_{h=1}^{H} \sum_{w=1}^{W} A_{k,hw}^+}$ |
| $\text{SegEPG}_k$ | $\dfrac{\sum_{h=1}^{H} \sum_{w=1}^{W} S_{k,hw} A_{k,hw}^+}{\sum_{h=1}^{H} \sum_{w=1}^{W} M_{k,hw} A_{k,hw}^+}$ |
| $\text{X-SegEPG}_k$ | $\dfrac{\sum_{h=1}^{H} \sum_{w=1}^{W} S_{k,hw} A_{k,hw}^+}{\sum_{h=1}^{H} \sum_{w=1}^{W} A_{k,hw}^+}$ |

Table 8: **Table comparing different version of the EPG metric**. Original EPG-metric (row 1, Wang et al. (2020)), Seg-EPG (row 2, Rao et al. (2023)), X-SegEPG (row 3, ours). Each EPG-metric is defined for a specific class $k$. The image is of dimensionality $H$, $W$, $M$ is the binary bounding box mask, $S$ is the binary segmentation mask, and $A^+$ is the mask for positive attributions.

## G   Correlation between EPG score and bounding box size

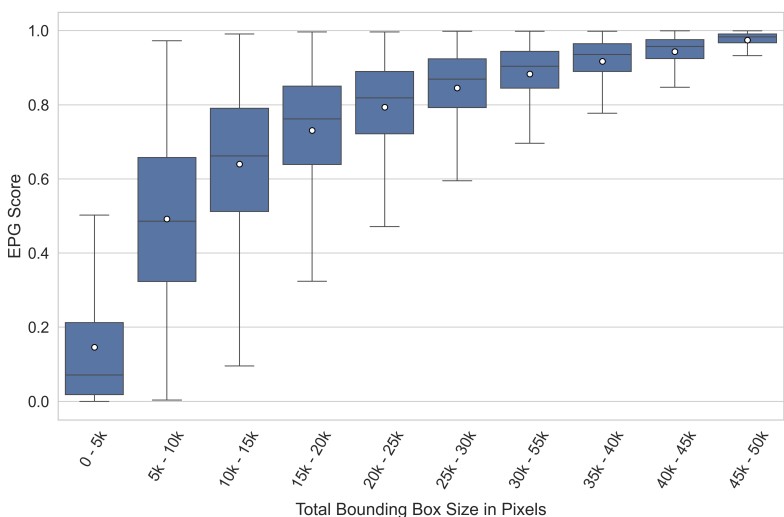

Figure 14: **Bounding Box Size vs EPG Score for a B-cos model**. The bounding box size corresponds to the number of pixels of the 224×224 image that is covered by one ore more bounding boxes of the same class. Each box represents the interquartile range (IQR) with the median indicated by a black line. The white dot within each box denotes the mean EPG score. The whiskers extend to the most extreme data points within 1.5 times the IQR.

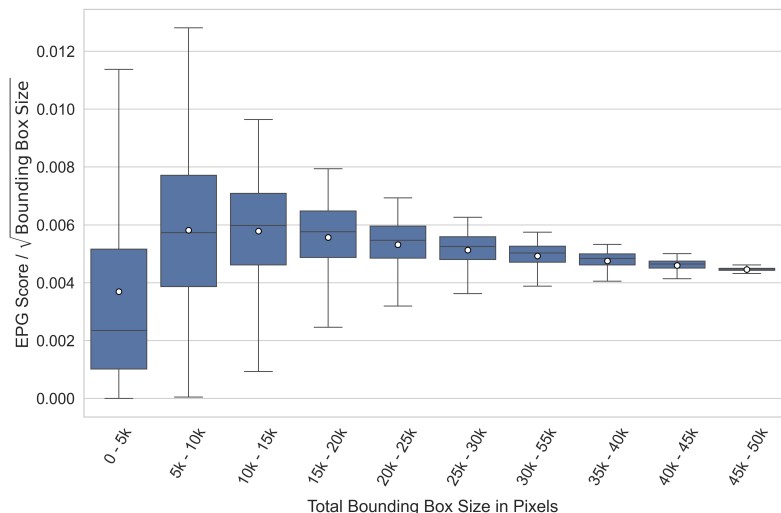

Figure 15: **Bounding Box Size vs normalized EPG Score for a B-cos model**. The bounding box size corresponds to the number of pixels of the 224×224 image that is covered by one ore more bounding boxes of the same class. Each box represents the interquartile range (IQR) with the median indicated by a black line. The white dot within each box denotes the mean EPG score. The whiskers extend to the most extreme data points within 1.5 times the IQR.

## H    Qualitative results: Waterbirds-100

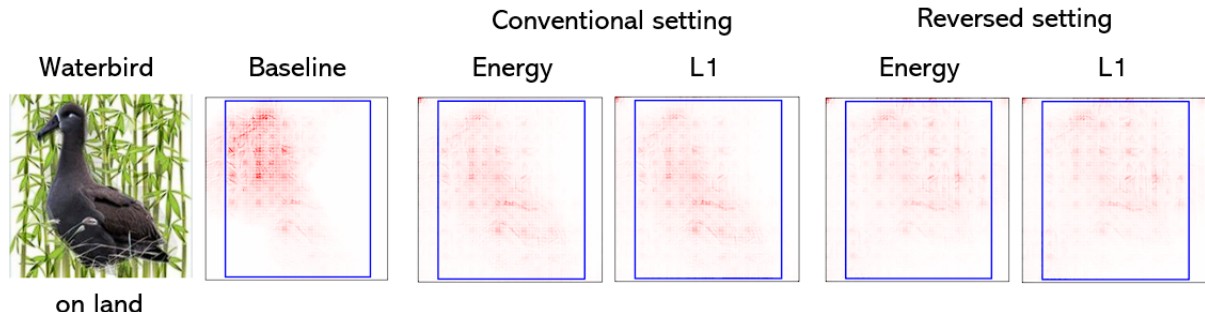

Figure 16: **Qualitative Waterbirds-100 result.** We randomly pick an image from the worst group (**G3**) to try and reproduce the setting from the paper. However, unlike in the original case, we do not observe a notable improvement. This could likely be due to random selection, so further qualitative analysis should be done to give a definite reproducibility assessment.

## I    Localisation performance based on IoU

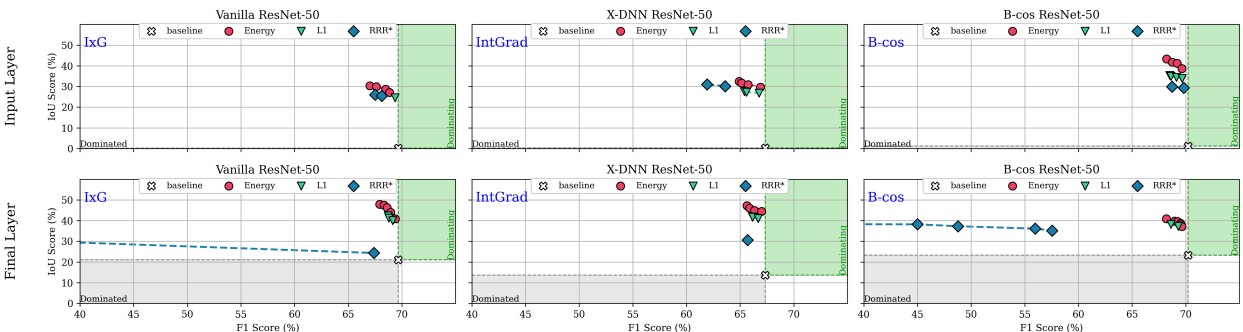

Figure 17: **IoU vs F1** on COCO for different localisation loss functions used during training (markers), models (columns) and model guidance layers (rows). For each configuration, best performing models on the validation set were evaluated on the test set and the Pareto fronts are plotted.

## J    Impact of context: Survey details

As mentioned in Section 3.5.3, we conducted a survey to examine the importance of context in image classification. This section provides the full details of the survey content, as well as some additional statistics that were not included in the main paper.

The survey consisted of six main questions and we collected a total of 110 responses. Each question would present the participants an image of either a real or a toy car. Initially, the participants were only shown a cropped version of the image focusing only on the object of interest, deliberately designed to exclude most contextual information. All six images are shown in Figure 18.

Upon seeing the cropped version of the image, the participants were asked to to try to identify whether the image depicted a real or a toy car (Qx.1). In addition, they were asked whether the predominant factor influencing their decision was the context, the object itself or both equally (Qx.2).

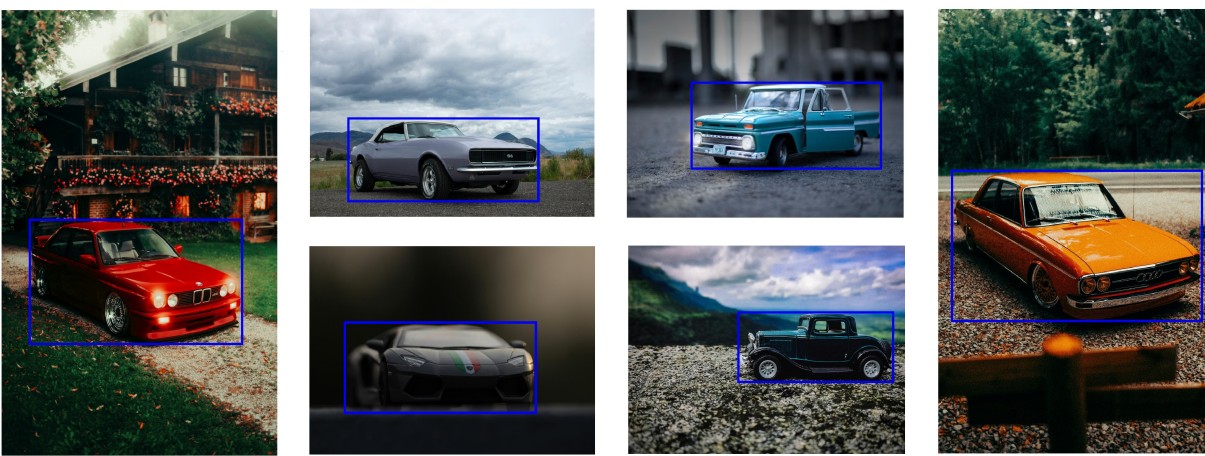

Figure 18: **Survey images.** Participants were first shown only the portion of the image within the bounding boxes (shown in blue) and the full images later

Following this, the participants were shown the full or original version of the image to evaluate whether access to additional context led to a change in their classification or the reasoning behind their decision (Qx.3 and Qx.4). Finally, the participants were asked to identify all the reasons they deemed relevant to their decision-making process (Qx.5). This question offered a more fine-grained selection of choices and allowed for multiple answers with the aim to get a deeper insight into what particular features of the object or the context influenced the decisions. A complete overview of the survey questions can be found in Table 9. Furthermore, Figure 19 shows the distribution of answers for each question.

Further statistical analysis of the results was carried out and two figures of particular interest were presented in Section 4.2.3. We applied appropriate statistical test in both instances, demonstrating statistical significance. The details are found in Table 10. Moreover, Fig 20 shows additional interesting statistics which were excluded from the main paper and for which we did not run statistical tests.

Table 9: **Survey questions and answer options.** The 'x' in the question number template e.g., **Qx.1** stands in place of the image number, as the given set of five questions was the same for each of the six images.

| No. | Question | Answer Options |
|---|---|---|
| **Qx.1** | Is this a picture of a real or a toy car? | Real, Toy, Unsure |
| **Qx.2** | Which factor did predominantly inform your decision? | Object, Context, Both equally, Unsure |
| **Qx.3** | Now does this picture feature a real or a toy car? | Real, Toy, Unsure |
| **Qx.4** | Now which factor did predominantly inform your decision? | Object, Context, Both equally, Unsure |
| **Qx.5** | What aspects of the image led to your decisions? | Context, Size, Colour, Object details, Image clarity, Texture, Other |

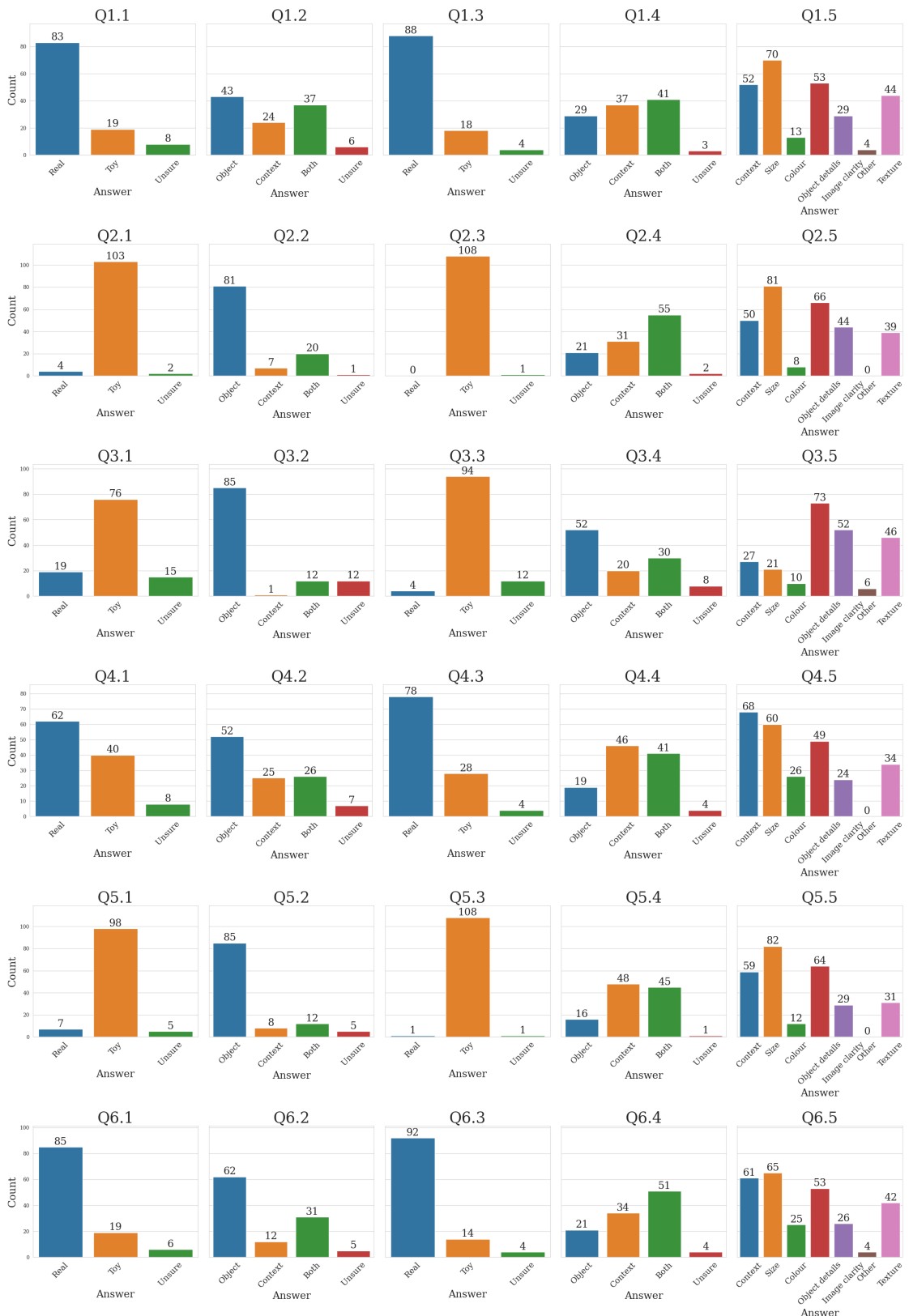

Figure 19: **Survey: answer distribution**

Table 10: **Summary of statistical tests on image accuracy and decision changes.** The paired t-test was used for Figure 9 and the McNemar's test for Figure 10

| Test | Statistic | P-Value | Significance level ($\alpha$) | Conclusion |
|------|-----------|---------|--------------------------------|------------|
| Paired t-test | $t = -4.42$ | 0.0069 | 0.05 | Significant change in accuracy after observing context |
| McNemar's test | $\chi^2 = 19.0$ | <0.0001 | 0.05 | Significant change in decisions after observing context |

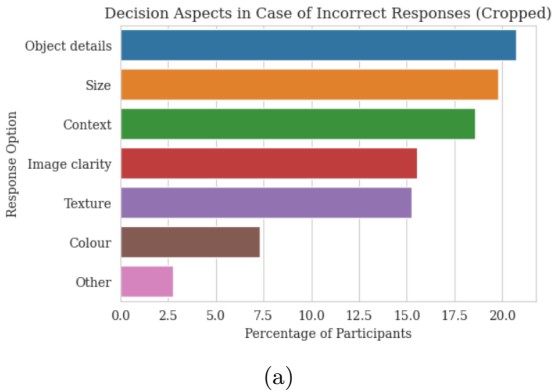 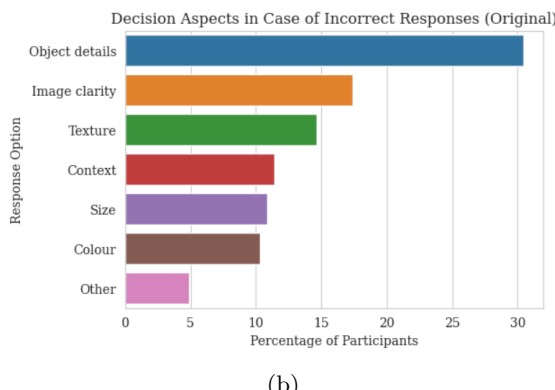

(a)            (b)

Figure 20: **Analysis of decision aspects in incorrect classification cases.** (a) Decision aspects when seeing only the bounding box information. (b) Decision aspects when seeing the full image. In both cases, when making incorrect classifications, participants focused mainly on on-object features.

## K   On-object localization performance

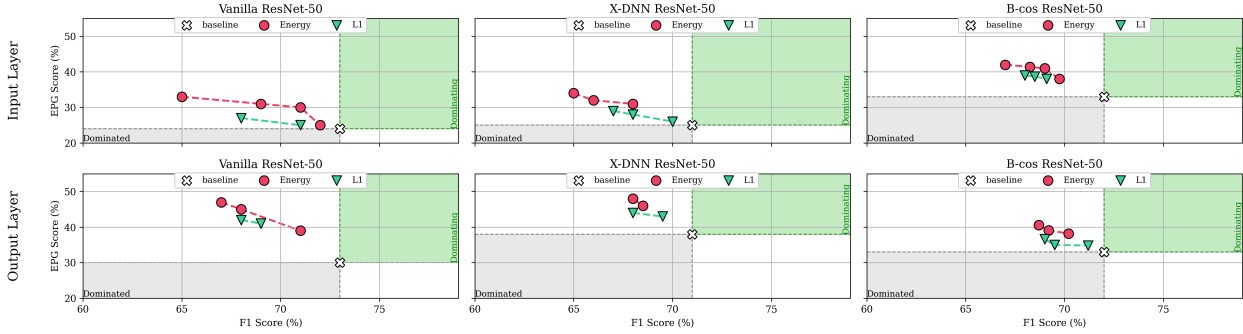

Figure 21: **On-object EPG results.** In our attempt to reproduce claim **R3** of Rao et al. (2023), we evaluate the on-object performance across models (columns) and layers (rows) for the Energy and $L_1$ localization loss. Following the original result, we find that Energy loss is more effective than the $L_1$ loss in localizing attributions to the object compared to the background regions within the bounding boxes. This is due to the formulation of the respective losses, where the $L_1$ loss promotes uniformity in attributions within the bounding box across both on-object and background features, whereas the Energy loss does not promote such a uniformity, leaving the free to decide which regions withing the bounding box are important for its decision.

