# OpenReview forum: "“Studying How to Efficiently and Effectively Guide Models with Explanations” - A Reproducibility Study"
_TMLR — Accepted by TMLR_

### Review · Reviewer_TmdR · 2024-03-13

**Summary Of Contributions:**

The authors focus on the reproducibility of Rao et al. (2023) regarding model guidance using attribution priors and extend the original research by addressing limitations in the Energy Pointing Game (EPG) metric, exploring the role of contextual information in object detection, and proposing improvements in model robustness and efficiency. Through thorough experimental validation and qualitative analysis, the paper substantiates the original findings, proposes corrections for EPG metric biases, and highlights the significance of contextual cues in certain scenarios.

**Audience:**

No

**Claims And Evidence:**

Yes

**Requested Changes:**

- Expanded Reproducibility Scope (if possible): Incorporating the claims not covered in the reproducibility study, specifically the effectiveness of the Energy loss in focusing on on-object features, is crucial. While computational resources are a limitation, seeking partnerships or leveraging cloud computing resources could enable these additional experiments. This would provide a more complete verification of the original work's claims and is critical to securing a recommendation for acceptance.

- Future Research Directions: Providing a more detailed discussion on future research directions, especially on how the proposed modifications could be integrated into existing model guidance frameworks or applied in other areas of AI, would elevate the contribution of the paper. Suggestions for how the community can address unanswered questions or the next steps in evolving model guidance practices would be beneficial.

**Strengths And Weaknesses:**

Strengths
- Comprehensive Reproducibility Study: The submission provides a detailed and thorough reproducibility assessment of the original work by Rao et al. (2023), confirming most of their claims and demonstrating a strong commitment to validating previous research findings.
- Valuable Extensions and Improvements: It introduces significant extensions to the original study, particularly addressing the limitations of the EPG metric and the bias towards larger bounding boxes, enhancing the robustness and interpretability of model guidance techniques.
- Practical Contributions and Open Source Code: By providing modifications to improve model guidance and sharing the code for reproducibility and further research, the submission makes a practical contribution to the field of explainable AI.

Weaknesses
- Limited Scope of Reproducibility: Although the paper covers a substantial portion of the original study, it does not reproduce every aspect, particularly the claim related to the Energy loss focusing best on on-object features due to computational constraints. This limitation slightly narrows the scope of the reproducibility study.
- Potential Bias in Survey Methodology: The survey conducted to assess the impact of contextual information on image classification could be subject to participant bias or limitations in sample size, potentially affecting the generalizability of its findings.
- Generalization of Findings: The generalizability of the proposed corrections and extensions to a wider range of models, datasets, and domains remains an area for further investigation, ensuring that the findings are applicable across diverse scenarios in explainable AI.

---

> ### Author Response · Authors · 2024-06-12
>
> Dear reviewer,
>
> We thank you for the thorough feedback. Below, we address the specific points raised in the review:
>
> Expanded reproducibility scope:
> - Reviewer comment: The paper does not reproduce every aspect, particularly the claim related to the Energy loss focusing best on on-object features.
> - Response: We acknowledge this limitation and have expanded the reproducibility scope to include the missing experiment (R3) regarding the effectiveness of the Energy loss in focusing on on-object features. This additional experiment can be found in Section 4.1 and Figure 20 (Appendix J) of the revised submission.
>
> Potential bias in survey methodology:
> - Reviewer comment: The survey conducted to assess the impact of contextual information on image classification could be subject to participant bias or limitations in sample size.
> - Response: We have noted this feedback and acknowledge the potential biases in our survey methodology. We discuss this in Section 5 together with other points for future work.
>
> Generalization of findings and future directions:
> - Reviewer comment: The generalizability of the proposed corrections and extensions to a wider range of models, datasets, and domains remains an area for further investigation.
> - Response: We agree that generalization is crucial for the broader applicability of our findings. As requested, we have expanded our discussion on future research directions in Section 5, including suggestions for applying our proposed modifications to various models, datasets, and domains.
>
> We hope that these revisions address the reviewer's concerns and improve the overall quality and comprehensiveness of our paper. Thank you for your constructive feedback and the opportunity to strengthen our work.

---

### Review · Reviewer_RVME · 2024-04-03

**Summary Of Contributions:**

This is a reproducibility study of the paper “Studying How to Efficiently and Effectively Guide Models with Explanations” by Rao et al (ICCV 2023) which addresses “model guidance” which trains a model to make predictions “for the right reasons” (e.g. not on spuriously correlated background attributes).  The Rao paper is a large scale study of model guidance using different attribution methods, localization losses and architectures.  This submission reproduces a number of results from the original paper and highlights some of the results that are hard to reproduce and finally proposes a few extensions over the original work.

**Audience:**

No

**Claims And Evidence:**

Yes

**Requested Changes:**

I list a few minor nits below:
* It might help to define an attribution mask with a picture
* Misspelling on page 3 last two lines: constrains -> constraints
* In Sec 3.5.1 in the definition of X-SegEPG, I would recommend defining this using equations in addition to prose
* Nit in Sec 3.6: footnote number should go after the punctuation (i.e. the period).

Have the authors considered consulting the segmentation literature regarding attribution losses?  For example, one can consider dice loss which is popular in the segmentation community, differentiable and invariant to box size…

**Strengths And Weaknesses:**

Strengths:
* Overall this is a good write up of the details of reproducing the Rao et al work and it is a nice contribution to have confirmation of what parts of the Rao paper are reproducible or not.  This submission is highly detailed and articulates what reproduction details are clearly specified by the authors, and what is not, in the original paper.
* There are also some novel insights, including on the relationship of the “EPG” metric to bounding box size and a proposed way to correct for this correlation.

Weaknesses:
* I would note that the Rao paper has 1 citation (as of April 2024) so arguably the source paper is not a super impactful choice to reproduce.
* Also I question to what extent this is an independent reproduction given that the authors of this study are (mostly) running the original code released by the original authors
* Finally some of the novel contributions claimed by the authors here do not seem to be used in a deep way nor are particularly novel
    * E.g. the X-SegEpG metric is proposed (but not used in the remainder of the submission as far as I can tell(?))
    * The use of context in image classification and detection are well studied and should be cited (e.g. “An Empirical Study of Context in Object Detection” by Divvala et al)

---

> ### Author Response · Authors · 2024-06-12
>
> Dear reviewer,
>
> We thank you for the thorough feedback. Below, we address the specific points raised in the review:
>
> Citations and impact of source paper:
> - Reviewer comment: The Rao paper has only 1 citation (as of April 2024), which raises a question about its impact and choice for reproduction.
> - Response: At the time we started our project, the paper by Rao was very recent, and its citation count was not indicative of its future impact. The MLRC recommended using papers from the latest conferences, and in light of the increased use of machine learning models in safety critical applications we felt this paper offered an interesting research direction to investigate.
>
> Independent reproduction:
> - Reviewer comment: The study relies heavily on the original code released by the authors, questioning the independence of the reproduction.
> - Response: Our intention was to check whether the results could be reproduced using the publicly available code and to identify any issues that might prevent the reported results. We did, indeed, find issues in a small number of experiments, which we highlighted in our submission.
>
> Use of context in image classification:
> - Reviewer comment: The use of context in image classification and detection are well studied and should be cited.
> - Response: We have added the citation to the relevant part of our submission. Thank you for the suggestion!
>
> Requested Changes:
>
> Definition of attribution mask:
> - Reviewer comment: It might help to define an attribution mask with a picture.
> - Response: We have clarified the definition of the attribution mask in the text (Section 3.5.1) and refer for visual illustration in Figure 1 with extended explanation in the caption.
>
> Typographical errors:
> - Reviewer comment: Various typographical errors were noted.
> - Response: We have corrected all the typographical errors mentioned. Thanks for pointing them out!
>
> Equation for X-SegEPG:
> - Reviewer comment: Define X-SegEPG using equations in addition to prose.
> - Response: We have added an equation for X-SegEPG in Section 3.5.1 and included a table with all three metric formulas in Appendix E, for easier comparison.
>
> Consulting segmentation literature:
> - Reviewer comment: Have the authors considered consulting the segmentation literature regarding attribution losses?
> - Response: While this is a very interesting addition to consider, we did not have the time to experiment with this aspect. However, we added it to our discussion of future research directions in Section 5 (requested by reviewer TmdR).
>
> We hope that these revisions address the reviewer's concerns and improve the overall quality and comprehensiveness of our paper. Thank you for your constructive feedback and the opportunity to strengthen our work.

---

### Review · Reviewer_CXm8 · 2024-05-29

**Summary Of Contributions:**

- The authors prove that the paper "Studying How to Efficiently and Effectively Guide Models with Explanations" by Rao et al. is, for the most part, reproducible.

- The authors propose a modification for the SegEPG metric, to also take into account the areas of the image outside of the bounding box, which they call X-SegEPG.

- The authors also propose to normalize the EPG score by the bounding box area, and show that this decorrelates the EPG score with the bounding box size.

- The authors conduct a study on 6 images, showing that, for human participants, context can be important.

- The authors compare the effect of training with sparse segmentation masks, vs training with sparse bounding box annotations, and vs training with full bounding box annotations.

**Audience:**

Yes

**Claims And Evidence:**

No

**Requested Changes:**

- Authors should provide clear evidence on why X-SegEPG is superior to SegEPG
- Authors should provide additional evidence showing why the EPG score normalization is needed.
- Authors should modify their claims of the human study unless they provide convincing evidence that machine learning models also have this limitation. This point is particularly important as it goes against the whole idea of model guidance and having the model be "right for the right reasons". The claims on this point introduce confusion on whether it is really possible for models to produce a correct label focusing on the object only. If the authors say it's not possible, they should show it.
- Authors should discuss the conclusions of their study on the effect of training with sparse/dense bounding box annotations vs sparse segmentations masks.

**Strengths And Weaknesses:**

This paper contains very new original material, as for the most part, it consists of a reproducibility study of a paper which already has open source code.

The proposed X-SegEPG metric makes sense conceptually, but the authors only motivate it with some toy examples shown in Fig. 1. No real results are shown to illustrate how X-SegEPG produces different results to SegEPG and why it is better.

The authors show that normalizing the EPG score by the bounding box area decorrelates it with the bounding box size. However, it is not clear why having a higher EPG score for larger bounding boxes is an issue.

The study conducted on 6 images with real human participants is arguably representative of how machine learning models can behave. While people can struggle differentiating a real from a toy car without context, a machine learning model might not have this limitation. I find this study and its conclusions to be misleading and contrary to the premise of model guidance.

The study of the effect of training with sparse/dense bounding box annotations vs sparse segmentations masks is not discussed at all, just presented in Fig 10. We don't know what are the conclusions of this study.

---

> ### Author Response · Authors · 2024-06-12
>
> Dear reviewer,
>
> We thank you for the thorough feedback. Below, we address the specific points raised in the review:
>
> Focus on reproducibility:
> - Reviewer comment: "...for the most part, [the paper] consists of a reproducibility study of a paper which already has open source code."
> - Response: We want to highlight that our submission is to MLRC, a journal dedicated to reproducibility studies, accepting submissions through TMLR for the first time this year. Thus, our work indeed mainly focuses on reproducibility, but we also introduce several important extensions to the original study.
>
> Requested changes:
>
> Evidence for X-SegEPG:
> - Reviewer comment: Provide clear evidence on why X-SegEPG is superior to SegEPG.
> - Response: We have added results for the three metrics (EPG, Seg-EPG, X-SegEPG), to motivate our introduction of this metric. We do note that evaluation is performed on a single model and a limited subset of the data due to computational constraints, but we hope it is sufficient to demonstrate why X-SegEPG is a more robust and representative metric. Additionally, we have rephrased the text in Section 3.5.1 to more clearly motivate our choice.
>
> Justification for EPG normalization:
> - Reviewer comment: Provide additional evidence showing why the EPG score normalization is needed.
> - Response: As the bounding box increases, the EPG is bound to increase by definition. When the bounding box is equal to the size of the image, EPG will have the maximum score of 1 regardless of attribution location. Larger bounding boxes make it easier to score higher in EPG due to more room for error, making high EPG values for larger boxes often too optimistic. This underscores the need for normalization to provide a more accurate assessment of model performance. We adapted Section 4.2 to more clearly convey this idea.
>
> Survey and context:
> - Reviewer comment: The claims on this point introduce confusion on whether it is really possible for models to produce a correct label focusing on the object only. If the authors say it's not possible, they should show it.
> - Response: Our aim was not to contradict the idea of being "right for the right reasons" but to reassess what constitutes "right reasons". Our survey examples aimed to show cases where context is not spurious, but necessary for accurate classification (at least for humans). We do not claim that machine learning models also have this limitation, but we rather want to pose the following question: If humans can decide when to use context to aid their decision or when to focus only on the object features, why should we constrain a model from doing so? We do recognize that our study involved a small number of images and participants, raising potential biases. Thus, we present this as an area for further robust investigation. We include this discussion in Section 5 and slightly rephrase our narrative in Section 4.2.
>
> Conclusions of the study of segmentation masks and sparse annotations:
> - Reviewer comment: Discuss the conclusions of the study on the effect of training with sparse/dense bounding box annotations vs. sparse segmentation masks.
> - Response: We apologize for the oversight and thank the reviewer for pointing this out. We have added the conclusions of this study in the revised submission in Section 4.2.
>
> We hope that these revisions address the reviewer's concerns and improve the overall quality and comprehensiveness of our paper. Thank you for your constructive feedback and the opportunity to strengthen our work.

---

### Decision · Action_Editor_YKrs · 2024-08-12

**Recommendation:** Accept with minor revision

**Comment:**

The paper is a reproducibility study of the work "Studying How to Efficiently and Effectively Guide Models with Explanations" by Rao et al, which conducted a comprehensive evaluation of model guidance for object classification, examining various loss functions (particularly the use of EPG as both a performance metric and a training loss), attribution methods and models to assess their effectiveness.

In addition, the paper proposes two main extensions: X-SegEPG, an enhancement to the metric SegEPG  used in the original work for evaluating the effectiveness of model guidance, and normalizing the EPG score by the bounding box area to eliminate the correlation between EPG score and bounding box size. A preference study involving 6 images is used to illustrate the significance of context for human understanding. Finally, they compared the impact of training models using sparse segmentation masks, sparse bounding box annotations, and full bounding box annotations to assess the robustness. Findings suggest that Energy loss inherently guides models to on-object features without the requirement for segmentation masks.

All reviewers agree that the paper successfully validated the reproducibility of the majority of the findings in the paper by Rao et al.

Two reviewers RVME and TmdR lean to accept the paper while Reviewer CXm8 recommends rejecting the work.
In their initial reviews, reviewers RVME and TmdR raised several points that the authors responded (and addressed) in detail. Both considered that their main concerns had been addressed by the authors.
Reviewer RVME suggested providing equations to explain the proposed X-SegEPG metric. The authors added clarifications addressing the reviewers point. I will further suggest to the authors to include a small section to explain the whole notation and procedure. As a reproducibility study, it makes sense that the work assumes readers are familiar with the original work and notation, but it would make it more readable (e.g. including the beginning of section 3 in the original paper as well as Figure 3, maybe in the appendix).

The main objections made by Reviewer CXm8 are:
- No significant results showing why X-SegEPG should be preferred.
- The EPG normalization is not needed
- Human survey. This is not representative of the behavior of a machine learning model.

In Rao et al, SegEPG is used to show that even if the Energy Loss uses bounding boxes to guide the explanations, most of the attributions fall within the objects themselves. The authors note that ignoring the attributions falling outside of the bounding box leads to over optimistic values. In their response the authors added some numerical experimentation to show that this is the case. I believe that the point raised by the authors makes theoretical sense and the new results show that this is the case in practice (quite significantly in these results). Having said that, reviewer CXm8 has a point that the paper could use some further analysis. Do any interpretations change qualitatively because of the use of this more accurate metric?

Regarding the normalization: adjusting the loss function (or metric) to account for object size is a common technique in segmentation algorithms. The authors further show that changing the loss function leads to improvements without affecting the F1 score. While the results are limited due to the experiments not being too comprehensive (due to computational constraints), it makes a point of its promise.

Finally, regarding the human study: the authors clarify in their response, that it wasn’t intended to explain how a machine learning model works but rather that humans use contextual information when solving classification tasks, which clarifies the point. I believe, however, that this finding is non-controversial: the classification of in-distribution images would likely benefit from using the context: real cars are more likely to be placed next to houses. From Rao et al “the guided model is discouraged from relying on contextual features, making the classification more challenging. [...] guidance can significantly improve performance when there is a distribution shift between training and test.” They do not assume that the “relevant pieces of information for image classification reside solely on the object of interest”. The point is that when moved to an out of distribution scenario, models (or humans) relying on context are likely going to make mistakes. The authors should amend this part of their work.

In sum: all reviewers agree that the paper successfully delivers the reproducibility study, which is the main purpose of the work. Two reviewers considered their concerns addressed and recommended accepting the paper. While reviewer CXm8 made some valid comments that I addressed above, two of the contributions proposed by the authors are sound and informative. Therefore I recommend accepting the paper. The authors should address the points mentioned above for the camera ready.

**Audience:**

The paper is clearly niche but, certainly of interest to a segment of the TMLR audience.

**Claims And Evidence:**

This is a reproducibility study. It proposes two extensions that are backed by evidence. The paper is limited by computational resources, but the experimentation is sufficient and properly done (e.g. using a complex dataset: COCO)

---

> ### Author Response · Authors · 2024-08-19
> **Revisions and Camera Ready Version**
>
> Dear Action Editor,
>
> We thank you for the time spent on the review process of our submission. We have just uploaded the camera-ready version. We have made the following modifications based on your comment:
>
> - We included Fig. 3 from the original paper describing the model guidance procedure in App. A
> - We extended App. F with notation details from Sec. 3 of the original paper
> - In Section 3.5.3, we rephrased our discussion on the human study to more clearly differentiate the role of context in out-of-distribution (OOD) versus in-distribution (ID) scenarios. Additionally, we slightly revised our hypothesis and conclusion to more accurately convey the main idea of our research question.
>
> We hope the above changes adequately address the requested minor revisions.
>
> Thank you once again for your valuable feedback.
>
> Best regards,
>
> The authors